# ACTIVE IN-CONTEXT LEARNING: ENHANCING THE GENERALIZATION OF LARGE MULTIMODAL MODELS

## ABSTRACT

The performance of Large Multimodal Models (LMMs) on downstream tasks improves substantially when examples of visual-text relationships are incorporated as context, with performance gains increasing as the number of examples and the context window size grow. However, collecting high-quality training sets for In-Context Learning (ICL) to retrieve multimodal examples is not trivial, particularly in specialized domains like healthcare, remote sensing, finance, and scientific research, due to the significant costs of manual labeling and strict privacy regulations. In this paper, we introduce Active In-Context Learning (AICL), a novel paradigm that eliminates the need for traditional training sets in multimodal ICL. AICL dynamically selects and annotates a small, highly informative set of samples in real-time during the query phase of LMMs. This active set evolves throughout querying, with the most relevant examples being continuously retrieved from it to optimize LMM performance on new data, without relying on pre-existing training sets. To construct an optimal active set, we propose Spectral-based Representative Sampling, which applies spectral clustering in the early query phase to select samples that are early, class-balanced, and representative, ensuring the active set captures key features of the data distribution and reduces data bias. To fully leverage the active set, we propose Similarity-enhanced TopK Prompt Construction, which retrieves the most relevant multimodal examples using a TopK similarity strategy and integrates the visual similarities between the multimodal examples and the query samples directly into the text prompts. By incorporating this similarity information, LMMs can better grasp the relationships, leading to more accurate and context-aware predictions. Experimental results on 10 specialized datasets and four LMMs show that our method significantly enhances LMMs' generalization performance. For example, in medical diagnosis tasks, our method, using only 10 annotated samples in the active set, outperforms existing ICL methods that rely on 2,000 annotated training samples.

## 1 INTRODUCTION

With advancements in large model research, Large Multimodal Models (LMMs) like GPT-4o (Achiam et al., 2023), Gemini (Reid et al., 2024), Claude (Anthropic, 2024), and Qwen2-VL (Wang et al., 2024) are becoming essential tools in both everyday tasks and professional settings, from interpreting medical images to enhancing virtual assistants. Recent studies (Zhang et al., 2023; Baldassini et al., 2024; Zhou et al., 2024) have demonstrated that LMMs can perform multimodal In-Context Learning (ICL) for supervised tasks without needing parameter updates. Unlike standard ICL (Mann et al., 2020), multimodal ICL relies on prepending multimodal demonstration examples, such as medical images with their class labels (Ferber et al., 2024), to the query, allowing LMMs to make context-aware predictions. These examples are typically retrieved from large training sets, such as MINT-1T (Awadalla et al., 2024), which includes 1 trillion text tokens and 3.4 billion images. As the size of LMMs context window grows, Agarwal et al. (2024); Jiang et al. (2024) report that adding more demonstration examples leads to obvious performance improvements in both generative and discriminative tasks, thus requiring even larger and more comprehensive training sets.

However, collecting large high-quality training sets for multimodal ICL in new tasks is challenging for three main reasons. First, the high cost of manual labeling presents a significant barrier, especially in specialized domains where the complexity of the data often requires domain expertise for

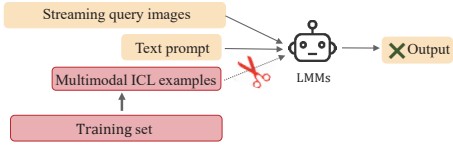 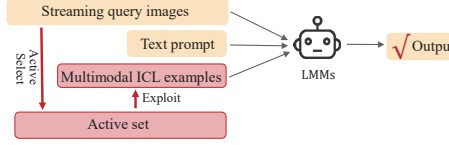

(a) Multimodal In-context Learning    (b) Active In-context Learning

Figure 1: Comparison of Multimodal In-Context Learning (ICL) and Active In-Context Learning (AICL): In specialized domains, acquiring high-quality multimodal training sets is often challenging. Differing from conventional ICL, AICL dynamically selects a small active set in real time during the LMM query phase to eliminate the need for pre-existing training sets.

accurate annotation (Sheng et al., 2008; Huang et al., 2015; 2021). For example, manual annotation of a pathology image can take anywhere from 13 to 48 minutes (Cazzaniga et al., 2024). Second, downstream query data often exhibit distributional shifts compared to training sets. This is particularly evident in healthcare, where variations in scanners and patient cohorts can exacerbate these discrepancies (Zhang et al., 2020; Guan & Liu, 2021). Zhou et al. (2024) has shown that such distributional shifts between ICL training sets and query data significantly reduce the performance of multimodal ICL in various discriminative tasks in healthcare. Furthermore, the real-time nature of downstream tasks, such as online systems, complicates the preparation of training sets due to continuous updates in streaming data (Rani et al., 2023; Sun et al., 2021). Third, privacy concerns severely limit the availability of training data. For example, the European Union's General Data Protection Regulation (GDPR) restricts access to patient datasets (Voigt & Von dem Bussche, 2017).

In this paper, we introduce Active In-Context Learning (AICL), a novel paradigm that enables dynamic multimodal ICL by constructing an evolving active set of annotated samples, which are curated by domain experts during query time. Unlike conventional ICL approaches that rely on extensive pre-annotated training sets, AICL selects and annotates the most relevant samples as the model encounters new data. This allows for the creation of a small but highly informative active set that adapts to the task on the fly (Fig. 1). AICL offers several advantages: (1) Enhanced sample efficiency, achieving superior performance with only 10 annotated samples, outperforming traditional methods that require 2,000 annotated training samples (Fig. 2). This is because, in real-world scenarios, training sets often exhibit distributional differences, providing limited contextual guidance for specialized query samples; (2) Rapid adaptation in new tasks, effectively gen-

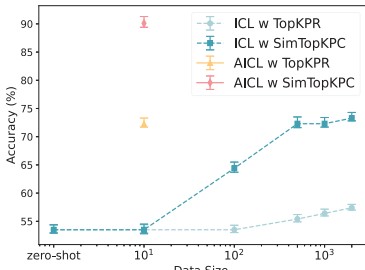

Figure 2: Comparison of ICL and AICL paradigms on the test set (Hos0 from the Camelyon17 dataset). AICL selects 10 samples directly from the test set to create the active set, while traditional ICL uses public datasets, specifically hos1-hos4, as the training set. However, the distribution of hos1-hos4 differs slightly from that of hos0.

eralizing with only a few valuable samples; and (3) Enhanced privacy protection, since it enables real-time annotation without accessing sensitive historical datasets. The key challenges AICL addresses include efficiently selecting the most informative samples and leveraging them in real-time ICL to enhance LMMs' generalization capabilities.

To construct an optimal active set, we propose Spectral-based Representative Sampling (SRS), which ensures that selected samples are early, class-balanced, and representative. (1) SRS performs early selection by identifying samples from the initial batch to ensure sufficient multimodal examples are available early in the ICL process, preventing learning delays (Fig. 5(a)); (2) SRS guarantees class balance by applying spectral clustering to high-dimensional features extracted from the streaming data, where it selects an equal number of samples from each cluster to ensure that all classes are well represented (Fig. 5(b)); (3) SRS ensures representativeness by choosing the samples closest to the cluster centers, capturing the core characteristics of each class (Fig. 5(c)). To fully leverage the active set, we introduce Similarity-enhanced TopK Prompt Construction (SimTopKPC). SimTopKPC uses a TopK Prompt Retrieval strategy to select the most semantically relevant multimodal examples based on feature similarities, and explicitly incorporates these similarities into the

text prompts. SimTopKPC enables LMMs to understand relationships between multimodal examples and the query samples. Empirical results highlight significant improvements, with a 30.1% accuracy increase in one-shot settings on the Camelyon17 dataset and a reduction in labeling costs by 90%, demonstrating the efficacy of our method (Fig. 3).

Our key contributions are: (1) We propose a novel learning paradigm, Active In-Context Learning (AICL), which enables training-set-free ICL by constructing an active set to improve the generalization capabilities of LMMs (§3.1). (2) We identify three critical characteristics for optimal sample selection in AICL: early selection, class balance, and representativeness. We propose the Spectral-based Representative Sampling and Similarity-enhanced TopK Prompt Construction modules to effectively select and leverage these minimal active samples (from §3.1 to §3.4). (3) We conduct comprehensive experiments on 10 benchmark datasets and four LMMs (Gemini 1.5 Flash, Gemini 1.5 Flash 8B, Qwen2-VL-72B, Glaude 3 Sonnet), which demonstrate that our approach significantly outperforms baselines in real-world scenarios (§4).

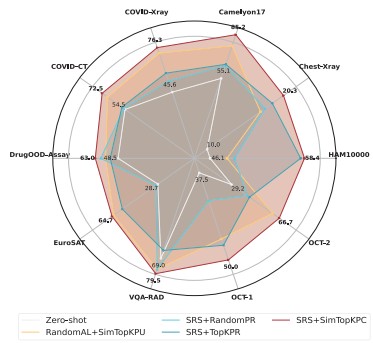

Figure 3: SRS with SimTopKPC consistently outperforms other ICL methods, demonstrating significant improvements in both generative and discriminative tasks.

## 2 RELATED WORK

### 2.1 IN-CONTEXT LEARNING

In-Context Learning (ICL) has emerged as a powerful strategy in natural language processing (Lu et al., 2021; Huang et al., 2024; Akyürek et al., 2024; Wies et al., 2024; Wei et al., 2022), driven by advancements in Large Language Models (LLMs) such as GPT-3 (Mann et al., 2020). For instance, ICL can use a set of review-sentiment examples to predict the sentiment of new reviews (Dong et al., 2022). ICL offers a more feasible solution compared to fine-tuning, which requires modifying model parameters—a process often impractical due to intellectual property restrictions or the computational expense of retraining large models. By contrast, ICL requires only examples from a relevant task, allowing LLMs to make predictions without altering the underlying model. This flexibility is particularly useful in specialized domains where data privacy and regulatory concerns prevent the free exchange of annotated datasets (Dong et al., 2022; Min et al., 2022).

Multimodal ICL has gained significant attention in the multimodal domain for its ability to enhance the generalization of Large Multimodal Models (LMMs) in specialized fields such as healthcare and visual interpretation (Zhou et al., 2024; Ferber et al., 2024). One key strength of multimodal ICL is its ability to retrieve demonstration examples from accessible training sets to improve performance. The pioneering Flamingo study (Alayrac et al., 2022) highlighted the crucial role of selecting high-quality examples for tasks like image generation and interpretation (Bar et al., 2022; Wang et al., 2023; Liu et al., 2021). However, collecting high-quality training sets remains a challenge. To address this, we introduce a paradigm for constructing an optimal active set during test time, enabling models to handle real-time queries without depending on pre-existing training sets.

### 2.2 ACTIVE LEARNING

Active Learning (AL) seeks to improve model performance by selectively labeling a small yet highly informative subset of samples (Michael, 2006; Settles, 2009). Traditional AL methods typically rely on metrics derived from model outputs, such as (1) Uncertainty (Siddiqui et al., 2020; Holub et al., 2008), including entropy; (2) Diversity (Brinker, 2003; Agarwal et al., 2020; Sener & Savarese, 2018), such as CoreSet; and (3) combined uncertainty and diversity strategies (Ash et al., 2019; Yang et al., 2015), such as BADGE. This paper seeks to enhance the generalization of LMMs at test time. Our approach differs in two key aspects: first, LMMs in our lightweight ICL framework often act as black-box models, making it difficult to extract conventional uncertainty or diversity metrics. As a result, we propose spectral-based representative sampling to construct an optimal active set.

Second, due to the complexity and size of LMMs, fine-tuning is often impractical, which is why we focus on developing the similarity-enhanced TopK prompt construction for more effective ICL example utilization, further boosting model performance without modifying underlying parameters.

# 3 METHOD

## 3.1 ACTIVE IN-CONTEXT LEARNING

Active In-Context Learning (AICL) aims to enhance the generalization of LMMs by constructing an 'active set' at test time, removing the need for a pre-existing training set. Let's start by clarifying the distinction between multimodal in-context learning and AICL.

**Multimodal In-context Learning** needs a pre-existing multimodal training set, $D_s = \{(x_n, y_n)\}_{n=1}^{N}$, comprising $N$ pairs (*e.g.,* an image and its label). Given a query image $x_q$ from a test set $D_q$, a text question $x_t$ (which could ask if the image belongs to a specific class or pose a free-form question about it) and a set of candidate answers $Y = \{y_1, y_2, ...y_m\}$ (where $Y$ could be class labels or a set of free-text phrases), multimodal in-context learning (ICL) is formulated as:

$$y_q = f(P, x_q, x_t) \mid y_q \in Y, \tag{1}$$

here, $P = \{(x_1, y_1), ..., (x_k, y_k) \in D_s\}$ denotes the multimodal ICL examples, chosen as $K$ samples from the training set, and $K \ll N$.

**Active In-Context Learning (AICL)** differs from multimodal ICL by building an 'active set' $D_l$ from streaming data batches, with the batch size represented as $b_z$. Instead of relying on a pre-existing training set, AICL selects ICL examples from this dynamically constructed active set. The expert labeling budget is defined as $B$, where $B \ll |D_q|$ and $B \ll |D_s|$, reflecting the high cost of expert annotation. Formally, when streaming data batch $D_q^i$ arrives in the $i$-th batch and $|D_l| < B$, indicating available annotation resources, AICL needs to efficiently select the most valuable active samples from $D_q^i$ for expert annotation. These samples, once annotated, are incorporated into the active set $D_l$. Subsequently, for a given query image $x_q^i$ within $D_q^i$, accompanied by a text question $x_t$, and a set of candidate answers $Y$, AICL is formulated as:

$$y_q = f(P^a, x_q^i, x_t) \mid y_q \in Y, \tag{2}$$

here, $P^a = \{(x_1, y_1), ..., (x_k, y_k) \in D_l\}$ denotes the multimodal ICL examples, chosen as $K$ samples from our active set.

In practice, once active samples are selected from $D_q^i$, and added to the active set, each query sample in $D_q^i$ is processed using this set to select ICL examples for prediction. Only after the entire batch is completed, the next batch $D_q^{i+1}$ is introduced. If the annotation budget has not been exhausted, active sample selection continues for $D_q^{i+1}$, and its samples are sequentially predicted; if the budget is depleted, predictions proceed directly using the existing active set. This cycle repeats until all batches are processed.

**Challenges.** According to the definition, AICL needs to address two critical factors that enhance the generalization of LMMs. **(1) Active selection:** Efficiently identifying informative active samples from streaming data batches and constructing the active set is crucial. Our experiments (Fig. 4(a)) show that selecting ICL samples using the TopK strategy from different active sets leads to varying performance gains, with performance differences between sets reaching up to 16.9% in downstream tasks. To enhance stability and performance, AICL requires effective sampling strategies to select optimal active samples. **(2) ICL example utilization:** Selecting and utilizing relevant ICL examples from the active set is equally important. Our experiments (Fig. 4(b)) demonstrate that random ICL example selections yield marginal benefits, with performance increasing only about 2% on datasets like EuroSAT and COVID-CT after annotating 10% of test samples. AICL must carefully select and leverage relevant ICL examples to significantly improve performance in the ICL process.

To tackle these challenges, we identify three key characteristics for optimal active samples: early selection, class balance, and representativeness. We introduce a Spectral-based Representative Sampling module to identify these samples and a Similarity-enhanced TopK Prompt Construction module to effectively select and utilize relevant ICL examples from the active set.

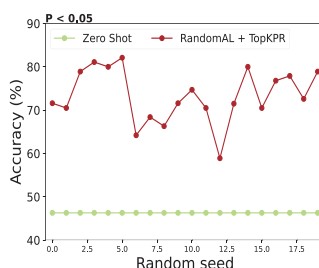 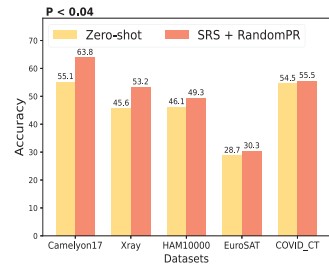

(a) Active sample selection is critical to AICL  (b) ICL examples utilization is critical to AICL

Figure 4: Challenges in AICL: (a) Selecting active samples with different random seeds (RandomAL) generates varying active sets. Using TopK ICL example retrieval from these sets results in performance variations of up to 16.9%. (b) Even though the active sets contain the most valuable samples selected by our SRS module, randomly selecting ICL examples (RandomPR) from these sets still yields minimal performance gains, with less than a 2% improvement observed in EuroSAT. In all figures, the notation '$p<0.05$' denotes statistical significance.

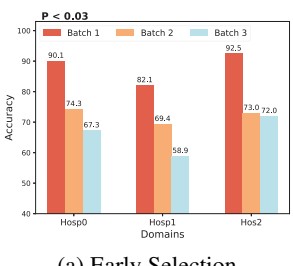 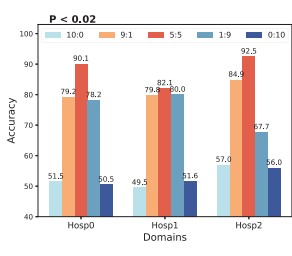 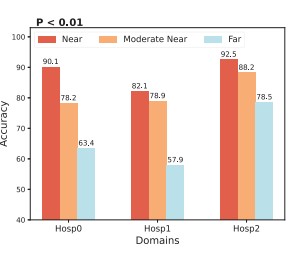

(a) Early Selection  (b) Class Balance  (c) Representative

Figure 5: Evaluation of the three traits of active samples using the first three domains in the Camelyon17 dataset follows two principles. First, Top-1 prompt retrieval effectively selects relevant ICL samples for each query. Second, these traits are interdependent, enhancing one another: (a) Early Selection: Choosing samples from the initial batch markedly improves performance, with 'Batch 1' being the earliest, followed by others sequentially. (b) Class Balance: Maintaining an even distribution of samples across classes, demonstrated by ratios such as '10:0' for normal and tumor classes, significantly boosts performance. (c) Representativeness: Selecting samples close to the class center, described as 'Near,' 'Far,' and 'Moderate Near,' also enhances performance.

## 3.2 THREE CHARACTERISTICS OF OPTIMAL ACTIVE SAMPLES

**Early Selection.** As shown in Fig. 5 (a), selecting active samples early significantly improves performance. Active samples chosen from the first batch outperformed those selected from the third batch by 22.8%, 23.2%, and 20.5% for different domains (hosp0, hosp1, and hosp2), respectively. This demonstrates the importance of beginning sample annotation as soon as the initial batch of data is received to develop a well-prepared active set ($D_l$). Delaying this step leads to poorer performance, as the early querying samples lack sufficient ICL examples for effective reference.

**Class Balance.** Fig. 5 (b) demonstrates that a balanced class distribution in the active set ($D_l$) greatly improves performance. In binary classification, the '5:5' ratio achieved a performance gain of 38.6%, 32.6%, and 35.5% over the extreme '10:0' ratio in different domains, respectively. This demonstrates the necessity of maintaining a balanced class distribution to prevent bias in the selection of ICL examples. When the active set is imbalanced, the model struggles to correctly identify underrepresented classes due to the absence of relevant ICL examples.

**Representativeness.** As shown in Fig. 5 (c), selecting samples closer to their class centroids significantly improves performance. Samples closer to the centroids outperformed those farther away by 26.7%, 24.2%, and 14.0% for different domains, respectively. This highlights the importance of selecting representative samples to maintain the quality of the active set. Centroid-proximate samples provide clearer class separation and more accurate information, enhancing the model's ability to interpret downstream queries and improving overall performance in LMMs.

Figure 6: The framework of our method combines Spectral-based Representative Sampling (SRS) and Similarity-enhanced TopK Prompt Construction (SimTopKPC). SRS leverages spectral clustering to select optimal samples near cluster centers in the initial streaming data batches, ensuring early, balanced, and representative sampling. SimTopKPC uses a TopK similarity strategy to retrieve relevant demonstration examples and embeds visual similarities between these examples and test samples directly into the text prompts, enhancing the understanding of LMMs.

### 3.3 SPECTRAL-BASED REPRESENTATIVE SAMPLING

As shown in Fig. 6, we introduce a Spectral-based Representative Sampling (SRS) module to explore the optimal active samples. The overarching formula is as follows.

$$D_l = \bigcup_{i=1}^{v} \left( \bigcup_{c=1}^{m} \left\{ \left( x_i^* = \operatorname*{argmin}_{x_i \in C_c} \|z(x_i) - \mu_c\|, y \right) \mid x_i \in D_q^i, |X_c| = n_c \right\} \right), \qquad (3)$$

Here, $v$ represents the total number of data batches with active sample selection, $D_q^i$ is the batch samples from the $i$-th batch, $z(x)$ represents the feature vector from the encoder such as CLIP-ViT, $y$ is the true label of $x_i$, $m$ denotes the number of cluster centroids, $\mu_c$ is the average features of cluster $c$, $C_c$ represents the samples within the cluster $c$, $X_c$ represents the number of active samples selected from the cluster $c$, where $|X_c| = n_c$. The label budget for the $i$-th streaming query batch $B^i = \sum_{n=1}^{m} n_c * m$, and the total label budget $B = \sum_{i=1}^{v} B^i$.

Next, we analyze why the selected $D_l$ meets the above three characteristics.

(1) To ensure a sufficient active set, SRS selects active samples from the initial streaming query batches $\{D_q^1, ... D_q^v\}$, where $v \ll n_{bz}$, with $n_{bz}$ representing the total number of batches.

(2) To obtain class-balanced active samples, we first need to cluster samples $D_q^i$ from the $i$-th streaming query batch to form well-defined groups. Spectral clustering (Von Luxburg, 2007) is applied to the features $z$ extracted by the pre-trained encoder from the $D_q^i$. This method is chosen due to its superior performance in handling high-dimensional, nonlinear feature spaces, outperforming traditional clustering algorithms and producing results more aligned with true class distributions, as shown in Fig. 7. Once clusters are established, an equitable selection of samples from each cluster is made to maintain class balance:

$$n_c = \frac{B^i}{m}, c \in 1, 2, ..., m, \qquad (4)$$

where $n_c$ represents the number of active samples selected from the $c$-th cluster, $B^i$ is the labeling budget in the $i$-th batch, and $m$ is the number of clusters formed.

(3) To obtain representative active samples, SRS builds on the well-formed clusters by selecting samples closest to each class centroid within each class:

$$x_i^* = \operatorname{argmin} \|z(x_i) - \mu_c\|, x_i \in C_c, \qquad (5)$$

where $x_i^*$ denotes the selected active sample from cluster $c$, $C_c$ represents the samples within cluster $c$, and $\mu_c$ is the average features (class centroid) of cluster $c$.

### 3.4 SIMILARITY-ENHANCED TOPK PROMPT CONSTRUCTION

Similarity-enhanced TopK Prompt Construction (SimTopKPC) aims to select and utilize the multimodal ICL examples from the active set, integrating both image and text modalities.

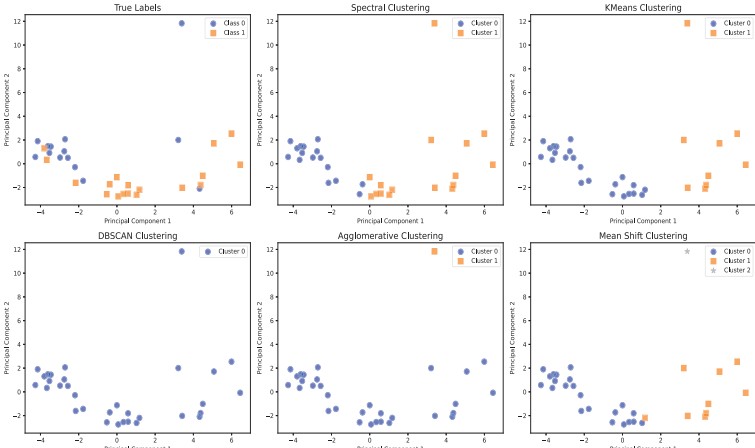

Figure 7: Comparison of clustering results from different unsupervised clustering algorithms on the hosp1 dataset at the first data batch. High-dimensional features, reduced via PCA, are plotted using the first two principal components on the horizontal and vertical axes. Among these, spectral clustering (top center) more closely matches the true labels (top left) than the other algorithms.

For the image modality, SimTopKPC employs the TopK unsupervised technique (Zhou et al., 2024) to select visual examples by assessing the similarity between the feature vectors of the query image $x_q$ and those within the active set. This similarity is quantified using the cosine metric, computed by a pre-trained vision encoder such as CLIP-ViT (Radford et al., 2021):

$$cos(x_q, x_a) = \frac{z(x_q) * z(x_a)}{||z(x_q)|| * ||z(x_a)||} , \qquad (6)$$

where $z(x)$ represents the feature vector from the encoder, with $x_q$ from the querying test set $D_q$ and $x_a$ from the active set $D_l$. The top $K$ images most similar to the query are selected by:

$$top_K(\{cos(x_q, x_a) : i = 1, ..., B\}) , \qquad (7)$$

where the operation $top_K$ selects indices corresponding to the $K$ most pertinent samples from the active set. These chosen images serve as in-context examples for LMMs, thereby enhancing their understanding and effectiveness in our specialized domains.

For the text modality, we refine the model's understanding of active labels in ICL samples by incorporating prompts that specify expert-designated true labels, such as: **'The image [i] has been accurately categorized as [true class] by domain experts.'** To further enhance the model's context-awareness of downstream query samples, we introduce prompts emphasizing the similarity between the ICL image and the query image, stating: **'Note that image [i] is similar to the next query image.'** By explicitly outlining these similarities, the model can better comprehend the relationships between ICL examples and query samples, allowing it to more efficiently identify key information and characteristics in new queries.

## 4 EXPERIMENTS

**Datasets Overview.** We focus on the task of specialized domains (medical imaging, remote sensing, and molecular imaging), where acquiring high-quality training sets is particularly challenging. Specifically, we conduct a rigorous evaluation involving discriminative and generative tasks and utilizing 10 diverse datasets: HAM10000 (Tschandl et al., 2018), Chest-Xray (Wang et al., 2017), Camelyon17 (Bandi et al., 2018), COVID-Xray (Han et al., 2021), COVID-CT (Yang et al., 2020), DrugOOD-Assay (Ji et al., 2022), EuroSAT (Helber et al., 2019), VQA-RAD (Lau et al., 2018), along with two proprietary optical coherence tomography (OCT) datasets. These datasets support various tasks such as multi-class classification, fine-grained classification, and visual question answering. Building on a related study (Han et al., 2024), we analyze practical test subsets from each dataset. As the two OCT datasets are proprietary, they are excluded from the pre-training data of

| Method | Camelyon17 | | | | | | EuroSAT |
|---|---|---|---|---|---|---|---|
| | Hosp0 | Hosp1 | Hosp2 | Hosp3 | Hosp4 | Avg | Acc |
| Zero-shot | 53.5 | 46.3 | 64.5 | 51.9 | 59.5 | 55.1 | 28.7 |
| RandomAL + RandomPR | 59.4 | 52.6 | 68.8 | 61.0 | 64.3 | 61.2 | 28.0 |
| RandomAL + SimTopKPC | 83.1 | 64.2 | 87.6 | 77.9 | 75.0 | 77.6 | 63.5 |
| BeginAL + SimTopKPC | 81.7 | 61.6 | **89.8** | 76.6 | 78.0 | 77.6 | 62.7 |
| SRS + RandomPR | 63.4 | 55.8 | 72.0 | 66.2 | 61.4 | 63.8 | 30.3 |
| SRS + TopKPR | 72.3 | 54.7 | 76.3 | 61.0 | 59.5 | 64.8 | 56.6 |
| SRS + SimTopKPC | **90.1** | **80.0** | 87.1 | **83.1** | **85.7** | **85.2** | **64.7** |

Table 1: Performance comparison on Camelyon17 and EuroSAT datasets

| Method | HAM10000 | | | | | COVID-CT | VQA-RAD |
|---|---|---|---|---|---|---|---|
| | RD | VMod | VMol | Vdis | Avg | Acc | Acc |
| Zero-shot | 26.1 | 33.8 | 55.4 | 68.9 | 46.1 | 54.5 | 69.0 |
| RandomAL + RandomPR | 30.9 | 38.9 | 53.0 | 77.8 | 50.5 | 57.5 | 60.1 |
| RandomAL + SimTopKPC | 33.1 | 32.5 | 37.8 | 71.1 | 48.3 | 68.3 | 76.5 |
| BeginAL + SimTopKPC | 33.4 | 32.1 | 62.1 | 81.1 | 52.2 | 68.8 | 76.5 |
| SRS + RandomPR | 32.1 | 38.9 | 53.0 | 73.3 | 49.3 | 55.5 | 53.5 |
| SRS + TopKPR | 35.2 | 44.6 | **65.1** | 86.7 | 57.9 | 57.0 | 63.5 |
| SRS + SimTopKPC | **36.4** | **46.5** | 63.5 | 86.7 | **58.4** | **72.5** | **79.5** |

Table 2: Performance comparison on HAM10000, COVID-CT, and VQA-RAD datasets

Large Multimodal Models (LMMs), resulting in distributional shifts. The inclusion of these datasets is intended to rigorously evaluate the proposed methods' effectiveness across varied distributions. A detailed description of all datasets can be found in the Appendix, Table 4.

**Models Overview.** We employ four state-of-the-art large multimodal models with publicly available API access: Gemini 1.5 Flash (Reid et al., 2024), Gemini 1.5 Flash 8B (which has a different number of parameters), (Reid et al., 2024), Claude3 Sonnet (Anthropic, 2024), and Qwen2-VL-72B (Wang et al., 2024). Given the free accessibility of Gemini 1.5 Flash, we primarily focus on its performance, with additional validation using the other models.

**Baselines.** In AICL, we design seven baselines: Zero-shot, RandomAL + RandomPR, RandomAL + SimTopKPC, BeginAL + SimTopKPC, SRS + RandomPR, SRS + TopKPR, and SRS + Sim-TopKPC. Here, RandomAL, BeginAL, and SRS represent active sample selection strategies, while RandomPR, TopKPR, and SimTopKPC denote various approaches for selecting and leveraging ICL examples from the active set. For active sample selection under the constraint of the label budget $B$, RandomAL selects and labels active samples using different random seeds in the initial batch ($v$ = 1) to construct the active set. We conduct experiments using five seeds (0, 1, 2, 3, 4) and report the averaged results. BeginAL selects the first $B$ active samples from the start of different data streams, with repeated experiments using the same seeds. SRS, our proposed strategy, identifies the top optimal samples. Notably, we exclude traditional active learning methods such as entropy (Holub et al., 2008) and coreset (Sener & Savarese, 2018), as LMMs, being API-based models, cannot reliably output uncertainty scores like entropy values. After constructing the active set, RandomPR randomly selects ICL examples from this set. In contrast, TopKPR selects ICL examples using a nearest neighbor approach (Eq. 7), while SimTopKPC applies the optimal selection and utilization strategy for ICL examples, developed in this work, across both image and text modalities. Implementation details are provided in the Appendix.

### 4.1 MAIN RESULTS

As shown in Tables 1, 2, and 3, we first evaluate the zero-shot performance of large multimodal models (LMMs) on eight benchmark datasets. These results reveal significant limitations in generalizing to specialized domains with zero-shot learning. In expert-dependent datasets like Camelyon17, EuroSAT, and HAM10000, zero-shot performance remains low at 55.1%, 28.7%, and 46.1%, respectively, highlighting the major challenges in applying LMMs to these domains.

| Method | Chest-Xray | | | COVID-Xray | | | DrugOOD Assay |
|---|---|---|---|---|---|---|---|
| | AP | PA | Avg | Sou | Tar | Avg | Acc |
| Zero-shot | 8.6 | 11.3 | 10.0 | 46.3 | 44.9 | 45.6 | 48.5 |
| RandomAL + RandomPR | 17.0 | 17.0 | 17.0 | 47.2 | 47.9 | 47.6 | 55.5 |
| RandomAL + SimTopKPC | 16.2 | 18.2 | 17.2 | 79.0 | 66.1 | 72.5 | 56.3 |
| BeginAL + SimTopKPC | 16.7 | 17.3 | 17.0 | 77.8 | 66.5 | 72.2 | 60.8 |
| SRS + RandomPR | 17.3 | 18.3 | 17.8 | 54.2 | 52.1 | 53.2 | 59.5 |
| SRS + TopKPR | 19.0 | 18.6 | 18.8 | 50.9 | 66.5 | 58.7 | 53.5 |
| SRS + SimTopKPC | **21.3** | **19.3** | **20.3** | **82.7** | **69.9** | **76.3** | **63.0** |

Table 3: Performance comparison on Chest-Xray, COVID-Xray, and DrugOOD Assay datasets

For active sample selection, the baseline RandomAL and BeginAL approaches, which select examples from the initial batch, are markedly less effective than our proposed SRS method. Even when integrated with SimTopKPC for in-context example selection and utilization, RandomAL and BeginAL fall short. Across the Camelyon17, EuroSAT, HAM10000, COVID-CT, NIH-Chest, Chest-Xray, and DrugOOD-Assay datasets, our SRS method outperforms zero-shot results by margins of 30.2%, 36%, 12.3%, 18%, 10.3%, 30.7%, and 14.5%, respectively, using only a 10% annotation budget. This substantial improvement highlights SRS's capability to pinpoint optimal samples from streaming data, far surpassing existing active learning baselines.

After constructing the active sample set using the SRS method, we evaluated the efficacy of RandomPR, TopKPR, and our SimTopKPC methods for selecting and utilizing in-context examples from the active set $D_l$ to enhance downstream task predictions. RandomPR, which randomly selects in-context examples, generally performs poorly. Conversely, combining SRS with a pre-trained vision encoder for example selection based on feature similarity—denoted as SRS+RandomPR—achieves significant improvements, with increases of 12.6% and 26.3% on the HAM10000 and EuroSAT datasets, respectively. However, this approach yields only marginal enhancements of 1.0% and 1.5% on the Camelyon17 and COVID-CT datasets, indicating inconsistent performance across different contexts. Our SRS+SimTopKPC strategy consistently surpasses all other methods, showing notable gains of 20.4% on Camelyon17 and 15.5% on COVID-CT compared to SRS+TopKPR. These results underscore the critical role of sophisticated visual active sample selection and contextual cues in AICL to improve the generalization capabilities of LMMs.

## 4.2 ANALYSIS

**Ablation on three characteristics.** To evaluate whether the active samples selected by our SRS achieve class balance and representativeness, we defined two metrics: Class Entropy (CE) and Class Similarity (CS). CE is calculated by the entropy (Holub et al., 2008) of $p$, where $p$ is the distribution vector of class quantities within the selected active samples, indicating class balance. A more uniform distribution vector, such as $p = [0.5, 0.5]$ in binary classification, signifies optimal class balance. CS quantifies representativeness as CS = Avg($s$), where $s$ measures the similarity of class-specific samples to their class centers, with higher averages suggesting closer proximity to class centers and greater representativeness, exemplified by $s = [1.0, 1.0]$ in binary classifications using cosine similarity. For analytical clarity, we compared the combined metric: Values = CE $\times$ CS across different tasks in the Camelyon and HAM10000 datasets, as depicted in Fig. 8, our SRS consistently achieves higher values, indicating superior selection of active samples that simultaneously meet criteria for class balance and representativeness. To maintain selection fairness, all baselines employed a priority-based selection for their first active sample batch at $v = 1$.

**Evaluation on Different LMMs.** We select two proprietary Large Multimodal Models (LMMs)—Gemini 1.5 Flash 8B and Claude 3 Sonnet—and an open-source LMM, Qwen2-VL-72B, for a comprehensive comparative analysis. Due to the high computational demands and associated costs of Claude 3 Sonnet, our experimental scope was confined to the Camelyon17 dataset. As depicted in Fig. 8(b), our method consistently demonstrates superior performance across various LMMs, proving its robust generalization capability across different domains and thereby expanding the application boundaries of generic large models.

**Evaluation on Distribution shifts.** We evaluate the robustness of our method in handling distribution shifts (the differences between the data used for training and the data encountered during testing)

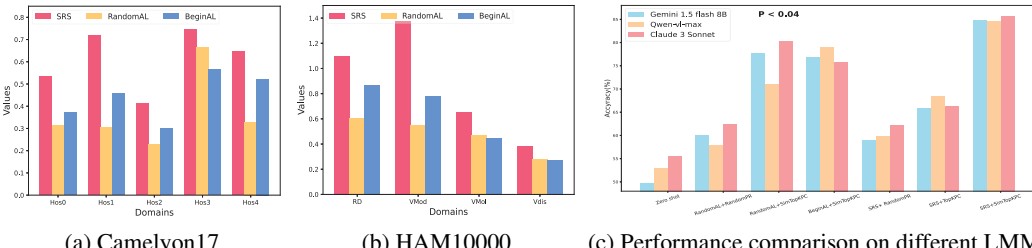

(a) Camelyon17        (b) HAM10000        (c) Performance comparison on different LMMs

Figure 8: (a) and (b) present the ablation study on Class Balance and Representativeness for the Camelyon17 and HAM10000 datasets. The horizontal axis shows different domains, while the vertical axis represents the composite metric (CE * CS), where CE measures class balance (entropy) and CS measures representativeness (average similarity to class centers). Higher values indicate better sample selection, highlighting the superior efficacy of our SRS. (c) shows the performance comparison across three LMMs.

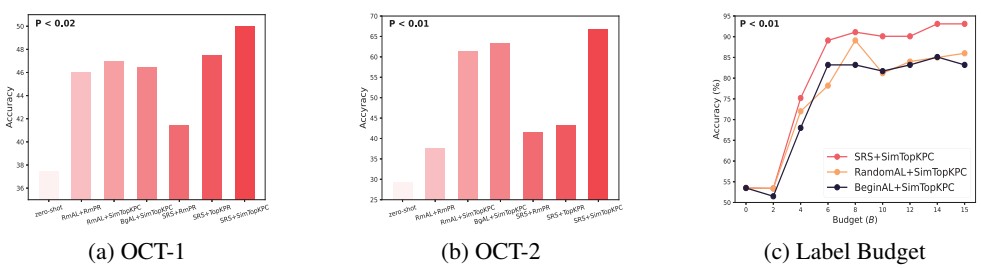

(a) OCT-1        (b) OCT-2        (c) Label Budget

Figure 9: Performance comparison on proprietary datasets with distribution shifts ((a) and (b)). Ablation study on label budget $B$ (c).

using two proprietary medical datasets (OCT-1 and OCT-2). These datasets were not included in the original training set of LMMs, allowing for an objective assessment of the models' ability to generalize across real-world distributional discrepancies. As demonstrated in Fig. 9(a) and (b), even with only 10% of the samples labeled, our method enables LMMs to effectively generalize across these distribution shifts, leading to more accurate diagnoses of eye diseases. By leveraging our approach, LMMs outperform existing baselines, demonstrating superior adaptability and robustness in handling real-world variations commonly found in medical imaging.

**Evaluation on Different Label Budgets** $B$**.** We analyze the impact of different label budgets $B$ on our method's performance compared to established baselines (RandomAL and BeginAL) within hosp0. As shown in Fig. 9(c), our approach (SRS + SimTopKPC) consistently outperforms these baselines across various labeling costs, due to its superior selection of optimal active samples.

## 5  CONCLUSION AND DISCUSSION

To address the challenge of acquiring high-quality training sets for In-context Learning, we introduced a novel paradigm, Active In-context Learning (AICL). AICL dynamically selects and labels informative samples to form an active set, thus eliminating the need for traditional training sets. We utilized Spectral-based Representative Sampling to construct this optimal active set, employing spectral clustering to ensure the samples are early, class-balanced, and representative. Our Similarity-enhanced TopK Prompt Construction module further refines the similarity relations between selected samples and query inputs in both image and text modalities, enhancing the model's understanding. This paradigm not only surpasses standard zero-shot performance with minimal annotation but also outperforms existing active learning baselines. Experimental results across multiple datasets and models confirm our method's enhanced generalization capabilities. This paper focuses on a relatively static scenario where the downstream data distribution remains unchanged over time. Future work will explore dynamic scenarios where the distribution evolves, enhancing the LMMs' ability to understand and adapt to new and changing tasks.

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

We present the following items in the Appendix:

A. Experimental Details.

B. Additional Experimental results.

# A EXPERIMENTAL DETAILS

## A.1 DATASETS

| Dataset | Task and image Type | # Classes | # Domain size | #Test set size | Per-domain label budget | Example Image |
|---|---|---|---|---|---|---|
| Camelyon17 | Tumor detection on pathology images | 2 | 5 | 450 | 10 | |
| HAM10000 | Skin disease classification on clinical photos | 7 | 4 | 450 | 10 | |
| COVID-Xray | Pneumonia detection on X-radiation | 3 | 2 | 450 | 20 | |
| Chest-Xray | Multi-label lung disease detection on chest X-rays | 15 | 2 | 400 | 20 | |
| COVID-CT | Pneumonia detection on Computed Tomography | 2 | 1 | 200 | 20 | |
| DrugOOD-Assay | Drug binding prediction on molecular images | 2 | 1 | 200 | 20 | |
| EuroSAT | Land use / land cover classification on satellite images | 10 | 1 | 300 | 30 | |
| VQA-RAD | Medical vision question answer with Yes/No | 2 | 1 | 200 | 20 | |
| OCT-1 | Ophthalmic Disease Diagnosis | 3 | 1 | 200 | 20 | |
| OCT-2 | Ophthalmic Disease Diagnosis | 4 | 1 | 200 | 20 | |

Table 4: Summary of benchmark datasets. We use 10 datasets spanning multiple domains (medical imagery, remote sensing, molecular imagery) and tasks (multi-class classification, fine-grained classification, and visual question answering).

We evaluate model performance on 10 datasets spanning multiple domains, including medical imaging, remote sensing, and molecular imaging, covering tasks such as multi-class classification, fine-grained classification, and visual question answering (VQA). For the VQA task, we specifically assess the model's accuracy on yes/no questions. The two proprietary OCT (OCT-1 and OCT-2) datasets facilitate the evaluation of the model's robustness in handling unseen downstream data with distributional shifts. The OCT-1 dataset was generated using a commercial 70 kHz spectral domain OCT system and comprises 160 images categorized as Normal, Age-related Macular Degeneration (AMD), and Diabetic Retinopathy (DR). The OCT-2 dataset was collected using the Heidelberg Spectralis OCT system, containing 400 images classified into Choroidal Neovascularization (CNV), Diabetic Macular Edema (DME), Drusen, and Normal. A detailed summary of the datasets used in this study is provided in Table 4.

Following existing works (Zhou et al., 2024; Jiang et al., 2024), we construct test sets from the original test splits, where available, to evaluate model performance. We sample the test sets from the original datasets without replacement. For Camelyon17, HAM10000, and COVID-Xray, we utilize publicly available subsets. For the remaining datasets, we select test sets based on the number of domains: for datasets with a single domain, 200 samples are randomly selected; for EuroSAT, due to its larger number of categories, 300 samples are chosen. Furthermore, for each dataset, we designate approximately 10% of samples per domain as active samples in our AICL. For example, in a domain with 100 samples, 10 are actively labeled, and in a domain with 200 samples, 20 are labeled, following this proportional approach.

## A.2 PROMPTS

Referencing the template designed for existing Prompts, the complete prompt for our paper is as follows:

```
image_descriptions = [f"Image {i+1} has been accurately categorized as {
    image_class}" for i by domain experts, (desc, image_class) in
    enumerate(source_images). "Note that image {i+1} is similar with the
    next query image."]

prompt = Given the images, answer the following question, using the
    specified format.{images_description}.

Question: What is the class of the next image? Choices: {class_names}.

Please respond with the following format for each image:
---BEGIN FORMAT TEMPLATE---
Answer Choice: [Your Answer Choice Here]
Confidence Score: [Your Numerical Prediction Confidence Score Here From 0
    To 1]
---END FORMAT TEMPLATE---

Do not deviate from the above format.
Repeat the format template for the answer.
```

## A.3 IMPLEMENTATION DETAILS

In our experimental setup, we employ the CLIP (vit-large-patch14-224-clip-laion2b) configuration to extract features from each dataset sample, which are subsequently used for clustering and the TopKPR process. In the VQA task, we leverage CLIP's text encoder to extract textual features from the 'question' field and combine these with image features to select the top-K ICL samples. We adjust the budget $B$ for active samples based on dataset size: for datasets containing fewer than 200 samples, the budget is set at 10; for those with more than 200 samples, it is increased to 20, ensuring that active samples represent approximately 10% of the total streaming query samples. Correspondingly, the batch size is established at 32 for smaller datasets and increased to 64 for larger datasets. The TopKPR strategy, as outlined in Eq. 7, is implemented to select ICL example samples with the parameter $K$ set to 1, motivated by the relatively low token count in the input examples which enhances computational efficiency and reduces costs. We explore a static scenario with consistent query distributions, using the initial batch as a representative sample to approximate the entire dataset. Thus, we set $v$ to 1 for adequate ICL referencing.

## A.4 LIMITATIONS.

Our experiment is constrained by two principal limitations: the lack of contextual learning capabilities in existing open-source models for medical datasets and the restrictive recognition of only pre-defined natural labels, which complicates the design of medically relevant prompts. Consequently, our experiments were solely conducted on private large multimodal model APIs, which limits the comprehensiveness of our results. Additionally, the selected datasets, both medical and proprietary, are relatively small, typically ranging between 100 and 400 samples. This limited data volume re-

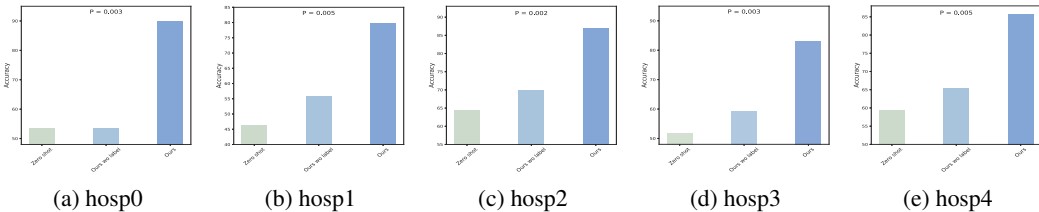

|  (a) hosp0  |  (b) hosp1  |  (c) hosp2  |  (d) hosp3  |  (e) hosp4  |

Figure 10: Comparison of performance on five domains (hosp0 - hosp4) in the Camelyon17 dataset across three settings: zero-shot without ICL examples, unsupervised ICL examples lacking correct labels, and our supervised ICL examples during the multimodal ICL process.

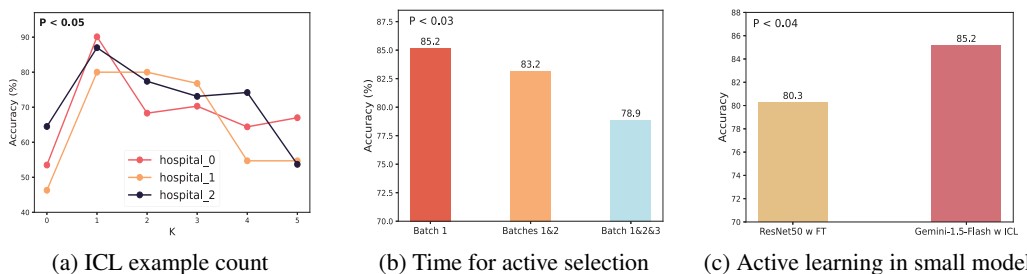

|  (a) ICL example count  |  (b) Time for active selection  |  (c) Active learning in small model  |

Figure 11: (a) Ablation study on label budget $B$. (b) Experimental results show that selecting all 10 active samples in the first batch outperforms distributing them across multiple batches (e.g., 5+5 in two batches ['Batch 1&2'] or 4+3+2 in three ['Batch 1&2&3']) (3) Comparison on active learning for the small model (ResNet50) and active in context learning for LMM (Gemini 1.5 Flash).

stricts the robustness and generalizability of our findings, emphasizing the need for further research with more diverse and extensive datasets to validate our conclusions.

## B    ADDITIONAL EXPERIMENTAL RESULTS.

### B.1    ICL EXAMPLES WITH VARIOUS SHOTS.

To examine the impact of in-context learning (ICL) examples within the SimTopKPC framework, we conducted an empirical study using the first three domains of the Camelyon17 dataset, adjusting the number $k$ in TopKPR from 1 to 5 per domain. Fig. 11(a) shows that optimal performance is attained at $k = 1$. Increasing $k$ reduces method efficacy due to the limited availability of actively labeled examples. An excessive number of in-context samples often leads to class mismatches between selected and query samples, where similar text prompts from SimTopKPC may be detrimental. In our setup, employing one in-context sample ($k = 1$) (Fig. 11(a)) optimally reduces the computational load on large multimodal models by minimizing extensive image token processing, thus conserving computational resources and prediction time.

### B.2    HYPERPARAMETER $v$ IN ACTIVE SAMPLE SELECTION.

In all experiments, we use the entire labeling budget ($B$) in the first batch $D_q^1$, with no updates to the active set in subsequent batches. This design is based on two key considerations: (i) In this paper, we focus on a static scenario, where the distribution of downstream query samples remains unchanged. As the first batch is typically considered representative of the overall dataset, selecting active samples from this batch approximates the full data stream distribution, providing sufficient ICL references. (ii) Experiments (Fig. 11(b)) show that allocating the full labeling budget to the first batch improves performance by 3% on average compared to dynamic updates, confirming that early selection provides adequate ICL support for the entire data stream. Delaying labeling reduces the model's ability to understand earlier batches due to the lack of sufficient ICL examples.

### B.3 Comparison on active learning for small models and active in context learning for LMMs.

We perform a comparative analysis of the application of active samples for fine-tuning small models versus their utilization in in-context learning to augment the generalization capabilities of large models. As depicted in Fig. 11(c), on the Camelyon17 dataset, the Gemini 1.5 Flash model, which employs in-context learning, achieves a performance improvement of approximately 5% over the gains observed in the ResNet50 model. This result validates the efficacy of our AICL paradigm, affirming its superiority in enhancing the generalization capabilities of large models compared to traditional updates of smaller models.

### B.4 Evaluation on Different Clustering Algorithms in active sampling

To evaluate the effectiveness of spectral clustering in our SRS module for active sample selection, we compare it with other clustering methods, including K-means, DBSCAN, Agglomerative, and Mean-Shift clustering on the HAM10000 dataset. All other conditions, including the SimTopKPC module, remain constant to ensure fairness. As shown in Table 5, our SRS with spectral clustering consistently outperforms the other algorithms. This demonstrates that spectral clustering, as applied in this study, yields more accurate initial clustering based on the original features, enabling the selection of the best active samples.

| Clustering | zero-shot | kmeans | agglometative | dbscan | meanshift | spectral (ours) |
|---|---|---|---|---|---|---|
| RD | 26.1 | 35.7 | 33.9 | 26.6 | 30.1 | **36.5** |
| VMod | 33.8 | 33.8 | 38.2 | 38.9 | 35.6 | **46.5** |
| VMol | 55.4 | 63.1 | 62.4 | 63.9 | 57.8 | **63.9** |
| Vdis | 68.9 | 82.2 | 82.2 | 77.8 | 73.3 | **86.7** |
| Avg | 46.1 | 54.2 | 52.3 | 51.8 | 49.2 | **58.4** |

Table 5: Performance comparison of clustering algorithms for active sampling on HAM10000.

