# OpenReview forum: "Active In-Context Learning: Enhancing the Generalization of Large Multimodal Models"
_ICLR.cc/2025/Conference — Submitted to ICLR 2025_

### Official Review · Reviewer_qXrr · 2024-10-26

**Soundness:** 2
**Presentation:** 2
**Contribution:** 1
**Rating:** 3
**Confidence:** 4

**Summary:**

Summary:
This paper explores Active In-Context Learning (AICL) for enhancing Large Multimodal Models (LMMs) in specialized domains. The authors propose to use Spectral-based Representative Sampling to construct an evolving active set from streaming data during query time, aiming to improve LMM performance without pre-existing training sets. A prompt construction approach based on similarity matching is proposed integrate selected examples into prompts to best leverage the active samples.

**Strengths:**

Strength:
1.I believe that exploring active learning in Large Multimodal Models (LMMs) is a meaningful and timely research direction, it offers potential to enhance LMM adaptability in specialized tasks.
2.I appreciate the idea of fully leveraging the active samples through prompt construction. It aligns well with the broader goal of active learning, which is to maximize annotation efficiency.

**Weaknesses:**

Weakness:
1.The problem setting is unclear. Critical details are missing.
The paper fundamentally lacks clarity on the distinctions between active learning data sampling, in-context learning, and the roles of training, unlabeled, and test data. Throughout, it remains ambiguous on how active learning operates within the in-context learning framework, providing little information on the processes behind data sampling, partitioning, or training. Essential details are missing, including which parts of the model are trained or fine-tuned and which loss functions, regularization, or optimization techniques are used. More specifically, the equations (Eq 1 and Eq 2) presented are merely prediction functions rather than “the learning process” claimed by the authors. Despite claims of “learning” from active sampling, the paper fails to specify which parameters are trainable or how the active samples are used to update the model. It remains unclear if the active learning component genuinely improves in-context learning or simply provides selective sampling during inference. This lack of detail on model trainability and learning mechanisms makes the method difficult to evaluate or replicate.
The lack of rigorous description of the problem setting eventually raise one of my major concerns: if portions of the test data are allowed to be sampled into the active set, while the remaining test data is used for evaluation. In this scenario, selective sampling of high-error test instances could lead to artificially lower test loss, even if the model's true performance has not improved. This potential bias would render the evaluation invalid.
Another critical issue with the paper’s problem setting is the frequent use of the term "test data", which conflicts with traditional interpretations and may lead to significant misunderstandings. Conventionally, test data is understood to have pre-existing labels (as it is used to evaluate the model’s performance) and should be excluded from the training set. However, the paper contains multiple confusing statements, such as "active sample directly select from test batch," which implies contradictory aspects 1). It suggests that all the test data are unlabeled initially, which would mean that if evaluation is to be conducted, all of them must be labeled; selectively labeling only part of the test data(i.e. Active learning) for evaluation would be inconsistent with this interpretation.2). It implies that part of the test data is being used for training, which is counterintuitive and contradicts the common practice.

2. Lack of novelty and justification for the choice of sampling strategies.
The paper subjectively dismisses traditional active learning methods, such as uncertainty and diversity sampling, claiming them to be unsuitable for the in context learning scenario.without providing concrete evidence or comparisons. A fair evaluation should include comparisons unless the authors can theoretically proof that they are unsuitable. In fact, the use of spectral clustering for sampling is itself a form of classical diversity sampling. It is not a novel methodology in active learning.  The authors do not demonstrate how it specifically outperforms other diversity-based techniques in identifying representative samples within the latent space.

**Questions:**

1.How is the "test data" defined and handled differently from conventional settings in this study?
2.What specific components of the model are trainable, and how does active sampling influence learning?

---

### Official Review · Reviewer_d3fN · 2024-11-01

**Soundness:** 2
**Presentation:** 3
**Contribution:** 2
**Rating:** 5
**Confidence:** 3

**Summary:**

In previous work, in-context learning (ICL) based on Large Multimodal Models (LMMs) has been proposed as an alternative to fine-tuning pre-trained models for various downstream tasks. ICL requires large (labeled) training sets related to the downstream task of interest. This is problematic because (1) labelling data points is expensive for many applications, and (2) the training set often exhibits distributional differences to the test set.

This paper proposes to (actively) select a very small subset of the test set to query labels for, which effectively solves both limitations above.

**Strengths:**

* The proposed method is simple and straightforward.
* The proposed method improves performance over several baseline methods on a number of different problems.
* Detailed and comprehensive experimental studies.
* The paper is well-written overall.

**Weaknesses:**

* The contributions are quite limited. The claimed novelties are (1) Run spectral clustering on current batch $D^i_q$ of test data points and construct active set by selecting $B/m$ data points close to each of the $m$ cluster centers, and (2) select K most relevant points from the active set to new query image using similarity information in latent feature space. In addition, they inform the text modality about true classes of data points in active set and about the similarity information between the query image and images in the active set. However, see points below.

* Using clustering to select representative and class-balanced points is a well known idea in active learning and goes back to e.g. [1]. In fact, it seems that this has already been done in the context of ICL (for LLMs, see [2]). I think the paper is currently lacking in terms of discussing prior works related to this. The start of Section 3.3 gives the impression that they are introducing this as a novel idea.

* In the clustering process, I assume you set the number of clusters to be the true number of classes. In general, this may not be known and could affect results of your method significantly.

* You introduce Eq. 3 which is suggesting to spread out selections for the active set over the $v$ batches. However, in your experiments you explicitly state that you consider a “static” setting where (as far as I understand) every batch is equally representative of the full test set. As a consequence, you exhaust the full budget $B$ in the first batch. Then, what is the point of introducing Eq. 3 (in its current form) if you do not evaluate it in your experiments (e.g., by considering a dynamic setting where the data distribution changes over time)?

* I am skeptical about the way you inform the text modality about similarity information (i.e., lines 358-365 of the paper). The first part (informing about classes of images in active set) is already what e.g. [4] does for ICL of LMMs in a non-active setting. I think you should cite them here. The second part (which I believe is the only novel part of the SimTopKPC component), is that you explicitly inform the model about the true class of the image which is most similar to the query image. Will this not simply result in the LMM predicting the class of the most similar image (from the active set) to the query image? In other words, is the LMM actually doing anything useful, or is it just predicting what you are explicitly telling it (which just results in a form of nearest-neighbor classification based on similarities in the latent space of the CLIP-ViT model)? You need to check this experimentally.

* The studies in [2, 3] address active ICL for LLMs. These seem highly relevant for your paper, but you do not cite them or discuss them in relation to your work. As mentioned, [2] also considers clustering to select representative samples.

* You talk a lot about “early selection” being important. This seems quite natural since you consider a “static” setting where each batch is equally representative of the test set. In a dynamic setting (which you do not even consider in your experiments), the idea of “early selection” would likely no longer be a good idea, since you have to adapt to a changing distribution of test points. In other words, you propose Eq. 3 (which effectively suggests a dynamic setting), but then recommend "early selection" which would probably not be a good recommendation for a dynamic setting.

* The way you explain the baseline methods is quite unclear currently. The difference between RandomAL and BeginAL is not clear. My assumption is that RandomAL means you construct the active set randomly, but this is not explicitly stated as far as I can tell. Also, why do you leave out certain combinations in the experiments, such as randomAL + TopKPR?

* I think the results presented in Figure 2 are a bit misleading. What if the training set is more representative of the test set, then perhaps far less data points are needed for equal performance. I understand the point you are making, but this example seems very extreme.


[1] Jaeho Kang, Kwang Ryel Ryu, Hyuk-Chul Kwon: Using Cluster-Based Sampling to Select Initial Training Set for Active Learning in Text Classification. PAKDD 2004: 384-388.

[2] Katerina Margatina, Timo Schick, Nikolaos Aletras, Jane Dwivedi-Yu: Active Learning Principles for In-Context Learning with Large Language Models. EMNLP (Findings) 2023: 5011-5034

[3] Yiming Zhang, Shi Feng, Chenhao Tan: Active Example Selection for In-Context Learning. EMNLP 2022: 9134-9148

[4] Guanglin Zhou, Zhongyi Han, Shiming Chen, Biwei Huang, Liming Zhu, Salman Khan, Xinbo Gao, Lina Yao: Adapting Large Multimodal Models to Distribution Shifts: The Role of In-Context Learning. CoRR abs/2405.12217 (2024)

**Questions:**

See points/questions under weaknesses.

Here are some additional minor things I found:

* Figure 10 is not referenced in the text.
* Figure 11, refers to figure c as 3 (in figure description).
* In Eq. 7, index $i$ is not used in the expression.
* Fig. 8 combines a score for representativeness and class balance. Can we conclude with full certainty that your method (which has the highest score) has the highest score for both representativeness and class balance, when you plot it this way?
* On line 297 you say $B_i = \sum_{n=1}^m n_c * m$, but it is just $B_i = n_c*m$ (from eq. 4).

---

### Official Review · Reviewer_nWWv · 2024-11-03

**Soundness:** 3
**Presentation:** 3
**Contribution:** 2
**Rating:** 3
**Confidence:** 3

**Summary:**

This paper introduces an active ICL approach for LMM. The method begins with spectral clustering, followed by querying unlabeled data points closest to each cluster centroid. In the test phase, a SimTopK technique is used to select relevant data from the query set to aid prediction. The authors conduct experiments to validate the proposed method's effectiveness and its comparative advantage over baselines.

**Strengths:**

The problem of active ICL for LMM is both practical and significant. This paper offers robust empirical studies that motivate the proposed selection criteria. The approach, which selects data based on proximity to centroids, is technically sound and well-motivated. The presentation is clear and well-organized, making the methodology easy to follow.

**Weaknesses:**

1) The technical contributions of this work appear limited. The proposed method relies on a conventional clustering-based selection strategy for forming the active set and querying through cosine similarity, which may lack originality compared to recent advancements in the field.
2) The datasets used for evaluation appear small and less widely recognized, which could impact the generalizability of the findings.
3) The choice of baselines is limited, with several relevant combinations, such as TopKPR and BeginAL, missing from the analysis. Furthermore, recent active ICL methods should be incorporated for a more comprehensive performance comparison.

**Questions:**

1)	The approach to querying data alongside the query set raises some questions. Why did the authors not consider using a pool-based active learning setting, where data could be queried in advance, prior to the test phase?
2)	In Figure 6, there appears to be a topKPR instance in the center of the figure. Could this be an error?

---

### Official Review · Reviewer_2CWW · 2024-11-04

**Soundness:** 3
**Presentation:** 3
**Contribution:** 2
**Rating:** 5
**Confidence:** 4

**Summary:**

The paper introduces a novel paradigm called Active In-Context Learning (AICL) that aims to enhance the generalization capabilities of Large Multimodal Models (LMMs) without relying on pre-existing training sets. AICL dynamically selects and annotates a small, informative set of samples at test time, evolving the active set to optimize LMM performance on new data.

The paper proposes Spectral-based Representative Sampling (SRS, that is, selecting an equal number of samples closest to each cluster center based on their proximity) to construct an optimal active set and Similarity-enhanced TopK Prompt Construction (SimTopKPC, that is, selecting the K samples closest to the input query from the active set as examples) to effectively leverage the active set.

Experiments on specialized datasets demonstrate significant improvements in LMMs' generalization performance with minimal annotation.

**Strengths:**

1. The paper is well-written and easy to follow.
2. The authors provide a comprehensive set of experiments across different datasets and models.
3. The figures and tables are well-organized, clear and easy to understand.
4. The method is relatively lightweight and easy to implement at the technical level.

**Weaknesses:**

1. Active In-Context Learning has been previously mentioned, referring to the paper "Active learning principles for in-context learning with large language models." by Katerina Margatina et al. (EMNLP 2023). The authors would do well to differentiate their approach from the existing work.
2. In terms of methodology, the authors employ rather common practices, such as spectral clustering and k-nearest neighbors, which do not exhibit significant innovation.
3. Experimentally, the methods compared by the authors are rather naive (such as random selection), and the results are consequently obvious; to be more convincing, the authors should introduce stronger baselines. They might consider incorporating more advanced selection algorithms from the field of active learning.
4. **Typos**:
    1. In Equation (4), the notation should be corrected from "$c \in 1,2,…,m$" to "$c = 1,2,…,m$".
    2. In Equation (7), the expression "$i=1,…,B$" should be changed to "$x_a \in D_l$", as the variable $i$ does not appear in the formula.
    3. The caption for Figure 11(a) is incorrect.

**Questions:**

1. On line 297, the definition of the label budget for the i-th streaming query batch is given as: $B^i = \sum_{n=1}^{m} n_c \cdot m$, but substituting Equation (4) into this formula results in $B^i = m \cdot B^i$?
2. Regarding the specific setup of $B^i$, is it equal for each batch? If not, how is it set?
3. Does the author select samples from the test set for annotation? If so, after the selection, the remaining test set will be different. How can the fairness of the validation be ensured?
4. The category space of the datasets used is not very large. Can the AICL framework be extended to larger label spaces? What are the potential bottlenecks?

---

### Meta-Review · Area_Chair_dJHD · 2024-12-20

**Metareview:**

The overall review of the paper is negative, and the authors did not provide a response during the rebuttal phase.

There are some weaknesses pointed by reviewers:

- Clarity of Problem Setting: The paper's problem setting is unclear due to missing critical details that distinguish active learning data sampling, in-context learning, and the roles of training, unlabeled, and test data. Moreover, there are unclear technical details, e.g., which parameters are trainable or how active samples update the model, making it difficult to evaluate or replicate the method.

- The paper claims novelty in its approach but fails to justify the choice of sampling strategies over traditional active learning methods like uncertainty and diversity sampling. The use of spectral clustering for sampling is not a novel approach in active learning, and the paper does not demonstrate its superiority over other diversity-based techniques.

- The paper does not cite or discuss relevant studies on active in-context learning for large language models, which is a significant oversight.

- The choice of baselines is limited, and the paper lacks a comprehensive performance comparison with more advanced selection algorithms from the field of active learning. Moreover, The paper's explanation of baseline methods is unclear, and certain combinations are missing from the experiments, which affects the interpretation of the results.

**Additional Comments On Reviewer Discussion:**

The authors did not provide a response during the rebuttal phase.

---

### Decision · Program_Chairs · 2025-01-22

Reject